# Application of machine learning with MALDI-TOF MS for rapid differentiation between methicillin-susceptible and methicillin-resistant *Staphylococcus aureus*

Yik-Shun Lin[1], River Chun-Wai Wong[2], Jiaxin Yu [iD][3], Kaichuang Yang[4], Leo Chun-Hei Wong[2], Ho-Fung Leung[1], Ingrid Yu-Ying Cheung[2], Viola Chi-Ying Chow[2], Ni Tien[5], Bang-Jau You[6], Christopher Koon-Chi Lai [iD][7]*, Margaret Ip[7]

**1** Faculty of Medicine, The Chinese University of Hong Kong, Hong Kong SAR, China, **2** Department of Microbiology, Prince of Wales Hospital, Hospital Authority, Hong Kong SAR, China, **3** ASTAI, Taichung City, Taiwan, **4** Department of Statistics, University of Oxford, Oxford, United Kingdom, **5** Department of Laboratory Medicine, China Medical University Hospital, Taichung City, Taiwan; Department of Medical Laboratory Science and Biotechnology, China Medical University, Taichung City, Taiwan, **6** Department of Chinese Pharmaceutical Sciences and Chinese Medicine Resources, China Medical University, Taichung, Taiwan, **7** Department of Microbiology, Faculty of Medicine, The Chinese University of Hong Kong, Hong Kong SAR, China

* chris.kclai@cuhk.edu.hk

## Abstract

### Background

Application of machine learning with matrix-assisted laser desorption/ionization time-of-flight (MALDI-TOF) mass spectrometry may allow rapid differentiation between methicillin-susceptible (MSSA) and methicillin-resistant *Staphylococcus aureus* (MRSA) and enable earlier AST-guided antibiotic use, but prior studies saw limited model performance. This study aims to apply novel machine learning techniques to a large dataset to create a prediction model with potential for clinical applications.

### Methods

This study has employed one of the largest datasets to date. 24487 *Staphylococcus aureus* isolates (13776 MRSA and 10711 MSSA) were collected between Jan 2021 and May 2024 in Hong Kong. These spectra were randomly divided into an 80:20 training-validation split to develop models of various structures. Top models, including a large-scale neural network (NN), the LightGBM gradient boosting framework (LGBM), and the weight-averaging ensemble model ("ensemble") of NN and LGBM, underwent prospective testing using 2975 additional clinical isolates (1867 MRSA and 1108 MSSA), and external validation using 1000 spectra (500 MRSA and 500 MSSA) from Taiwan.

**Data availability statement:** The base and external dataset used in this study is available on Kaggle at https://doi.org/10.34740/kaggle/ds/8530935. The prospective dataset is not available because only model outputs and performance metrics are captured. All code used for model fitting, evaluation and plotting, along with all model outputs, performance metrics and plots are available on a GitHub repository at https://github.com/maldi-tof-hk/maldi-tof-hk (Archived at https://doi.org/10.5281/zenodo.17395596).

**Funding:** The author(s) received no specific funding for this work.

**Competing interests:** The authors have declared that no competing interests exist.

## Results

The NN, LGBM, and ensemble models all achieved high performance with accuracy of 0.9284-0.9388 and AUPRC of 0.9843-0.9866 during prospective testing. The models are well-calibrated and confidence thresholds increased the accuracy to 0.9697-0.9777 by rejecting 20% of low-confidence predictions. External validation revealed accuracy of 0.695-0.723 and AUPRC of 0.8409-0.8765 with an increased number of false negatives. Shapley additive explanations revealed top feature groups consistent with previous studies, but feature importance was found to be geographically specific.

## Conclusions

We present new machine learning models with high performance in differentiating between MRSA and MSSA. Model performance can be further boosted with confidence thresholds, but models are not generalizable across different geographical areas. Clinical applications should use geographically specific models with fallback to traditional AST methods for low confidence predictions.

### Author summary

*Staphylococcus aureus* is a common type of bacteria that infects humans. It has a variant called Methicillin-resistant *Staphylococcus aureus* (MRSA) that is of high priority because it is resistant to multiple antibiotics. It is important to identify MRSA early so that appropriate antibiotics targeting the bacteria can be used. Currently, traditional methods to identify MRSA take 24 hours to incubate a sample of the bacteria and determine whether certain antibiotics are effective against it. However, it has been shown that data from routine mass spectrometry can be analyzed with machine learning to predict whether a sample may be MRSA before the incubation is performed. We have gathered more than 25,000 *Staphylococcus aureus* samples to build such machine learning models and have shown that the models achieve very high accuracy in local testing, but are less accurate when they are used to analyze foreign samples that are not covered in their training process. This means the models have the potential to encourage earlier use of appropriate antibiotics, but safe application of the models involves local usage and interpreting the models' own confidence on their predictions.

## Introduction

Methicillin-resistant *Staphylococcus aureus* (MRSA) is categorized as high-priority on the 2024 WHO bacterial priority pathogens list and poses a significant burden to the global healthcare system [1]. It can be identified by matrix-assisted laser desorption/ionization time-of-flight (MALDI-TOF) mass spectrometry (MS) and reported as *Staphylococcus aureus*, while antimicrobial susceptibility testing (AST) is needed to

determine whether it is MRSA or methicillin-susceptible *Staphylococcus aureus* (MSSA), and such a method requires an additional 24 hours of incubation.

Early administration of proper antibiotics is crucial in managing infections caused by these organisms, but rapid resistance profiling is required to achieve this goal. Current conventional culture-based methods take up to 48 hours to differentiate between MRSA and MSSA after sample processing, which prolongs the use of broad-spectrum antibiotics, further driving the development of antimicrobial resistance. Recent research studies have revealed the potential of MALDI-TOF beyond organism identification and suggest that prediction of resistance traits can be achieved by analyzing the mass spectra of *Staphylococcus aureus*.

There have been many attempts at resistance profiling in the past decade, mostly centered around non-deep learning approaches [2–13]. Although these models have shown initial success in identifying antibiotic resistance, most have yet to reach the performance required for clinical application. Deep learning approaches have been employed to a limited extent. For instance, Weis *et al.* adopted a multilayer perceptron (MLP) model to differentiate resistant and non-resistant strains of *Klebsiella pneumoniae* [14]. However, relatively small model size and sample size are the general limitations of these attempts.

The diversity of organism strains represents another major challenge in analyzing the spectra for the detection of antibiotic resistance. Strains from various geographical regions may have different phenotypic and/or genotypic characteristics, resulting in dissimilar patterns in their antibiograms as well as mass spectra. These changes in patterns are often unpredictable and intrinsically not generalizable, thus capturing them under a sample size constraint proves to be difficult. Limiting the scope of sample collection and model application may potentially enhance model accuracy by focusing on features of local strains.

In this study, we developed a composite model to classify MALDI-TOF spectra into MSSA and MRSA. A former study performed by the Yu et al. group used a dataset of 20359 clinical isolates to build their model for rapid differentiation between MSSA and MRSA [3]. With reference to their work, we built our machine learning models with one of the largest datasets to date, with 24487 clinical isolates for training and validation, and 3975 clinical isolates for 2 phases of testing. Our large dataset and multiple optimization strategies produce a model with consistently high precision and recall, rendering it a reliable tool to guide antibiotic use. By providing rapid and accurate resistance identification, our approach has the potential to improve treatment outcomes and contribute to the antibiotic stewardship program.

## Methods

### Study design

Fig 1 illustrates the various phases of this study. Three sets of *Staphylococcus aureus* mass spectrum data were used in this study, including the base dataset, the prospective dataset, and the external dataset. The base dataset was randomly split into a training subset and a validation subset in an 80:20 ratio. All models were first trained with the training subset and assessed with the validation subset. The top-performing models were then recruited into the prospective phase, in which data preprocessing pipelines and model parameters were frozen before the models were used to predict resistance in prospectively collected samples. To assess the geographical generalizability of the models, a retrospective external dataset from Taiwan was used in the final phase to evaluate the top-performing models.

### Bacterial isolates

For the base dataset, a total of 24487 *Staphylococcus aureus* mass spectra generated by two Bruker MALDI-TOF analyzers with MBT Compass IVD version 4.2.90 and 4.2.100 were retrospectively retrieved in the Department of Microbiology, Prince of Wales Hospital, Hong Kong. These strains were collected in the period between January 2021 and May 2024 and originated from patients in two independent acute hospitals in Hong Kong—one with 2,000 acute beds and the other

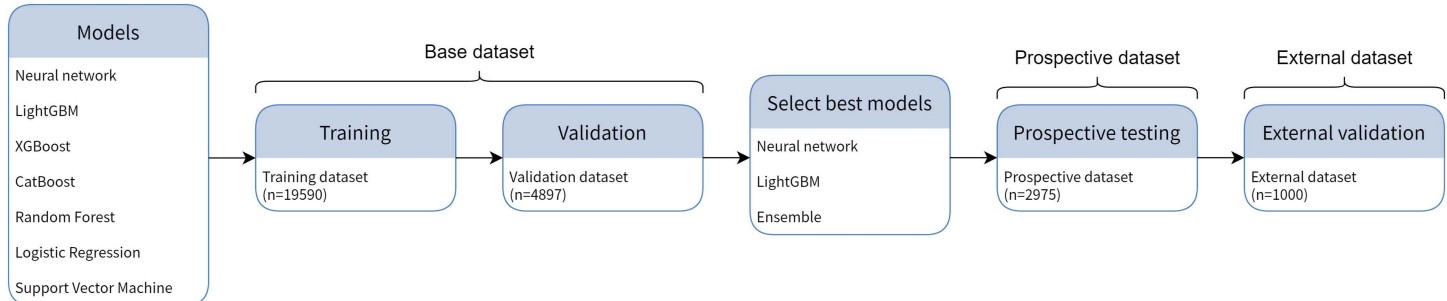

**Fig 1. Illustration of the model development and performance evaluation procedures of this study. All model types underwent hyperparameter tuning and 5-fold cross-validation.** Models with top performance in cross-validation were selected to undergo prospective testing and external validation.

with 700 beds—collectively serving a population of approximately one million. The strains were isolated from various types of clinical specimens (including wound swabs, sputum, blood culture, urine, tissue, pus, and miscellaneous MRSA/MSSA screening). This dataset contains 10711 MSSA and 13776 MRSA samples (S1 Table).

For the prospective dataset, 2975 local MALDI-TOF spectra were generated prospectively from clinical specimens using the same instruments and protocols. These specimens were retrieved from clinical samples of *Staphylococcus aureus* collected in the period between 1 June 2024 and 30 September 2024. This dataset contains 1108 MSSA and 1876 MRSA samples.

For the external validation, a total of 1000 *Staphylococcus aureus* mass spectra generated by a Bruker MALDI-TOF analyzer with MBT Compass version 4.1 were retrospectively retrieved at the China Medical University Hospital clinical microbiology laboratory in Taiwan, a location geographically distinct from Hong Kong. These samples were collected in the period between January 2020 and June 2021 and were isolated from blood, cerebrospinal fluid, ascitic fluid, pleural fluid, sputum, tissue, bile, stool, urine and other clinical specimens. This dataset contains 500 MSSA and 500 MRSA samples.

## Spectrum data

MALDI-TOF spectra data were retrieved from Bruker MALDI-TOF analyzers, whereas duplicate spectra from the same isolate were excluded. *Staphylococcus aureus* spectra were selected for subsequent analysis if the identification score was ≥ 2.0 with high consistency of category A, which represents the first and the second-best identification results that were matched in both genus and species levels. These raw spectral data were extracted using an in-house-developed Python pipeline. In brief, the nmrglue package was used to retrieve mass calibration constants and the intensity profile of each spectrum. The corresponding m/z profile was calculated using equations described by Titulaer *et al.* based on the retrieved mass calibration constants [15]. To ensure unbiased model validation, deduplication of spectra from the same patient was performed based on the hospital number within the same admission episode. As a result, MALDI-TOF spectra from the same patient or episode do not duplicate in both the training and validation sets.

## Data processing

After the acquisition of raw spectral data in the form of m/z values with corresponding intensities, data preprocessing was performed in accordance with the steps proposed by Weis *et al.* [14] A brief outline of the procedure is as follows:

1. Apply a square root transformation to all intensities to reduce variance.

2. Apply smoothing using the Savitzky-Golay algorithm with a half-window size of 10.

3. Perform baseline removal using 20 iterations of the SNIP algorithm.

4. Normalize intensities by setting the Total Ion Current to 1.

5. Trim the spectra to the range of m/z 2,000–20,000.

6. Partition data into 6000 discrete bins, each with a size of 3 m/z.

To facilitate data processing internally, data processing was implemented in Python using packages including Pandas, NumPy, SciPy, and pybaselines.

## Model construction

In this study, we investigated the performance of multiple machine learning models, including a densely-connected neural network, LightGBM [16], XGBoost [17], CatBoost [18], random forest, logistic regression, and support vector machine, as well as ensemble models using the above-mentioned models as constituents. With reference to the hyperparameters used in the study by Weis *et al.*, [14] they were optimized for each model type to minimize validation loss in the training dataset. Class balancing or class weighting in loss function was not implemented due to the relatively small class imbalance in the dataset (S2 Table) and the use of evaluation metrics not biased by class imbalance. A notable change in the process was the switch from Scikit-learn to Tensorflow Keras for the densely connected neural network to take advantage of hardware acceleration. Ensemble models were built by combining the best-performing model types with a weighted average of their outputs. They were then treated as separate model types and assessed independently. 5-fold cross-validation was performed for all model types to ensure stable performance. The optimal model was determined using performance metrics averaged over 5 folds.

## Performance evaluation

The base dataset was randomly split between training and validation sets in an 80:20 ratio, with 19590 training samples and 4897 validation samples. Model performance was assessed on each dataset using binary cross-entropy loss, accuracy, precision, recall, F1 score, and area under the precision-recall curve. The model outputs a continuous value between 0 and 1, while target values are discrete, with 0 indicating MSSA and 1 indicating MRSA. The numerical difference between the model outputs and the targets is quantified using binary cross-entropy, which is a logarithmic loss function that penalizes "confidently wrong" predictions.

When model outputs are interpreted as discrete classes, accuracy, precision, recall, and F1 score are used to assess performance. Their definitions are shown below:

$$Accuracy = \frac{TP + TN}{TP + TN + FP + FN}$$

$$Precision = \frac{TP}{TP + FP}$$

$$Recall = \frac{TP}{TP + FN}$$

$$F1\ score = \frac{2 \times Precision \times Recall}{Precision + Recall}$$

TP = True Positive (Model predicts MRSA while AST confirms MRSA), TN = True Negative (Model predicts MSSA while AST confirms MSSA), FP = False Positive (Model predicts MRSA while AST reports MSSA), FN = False Negative (Model predicts MSSA while AST reports MRSA).

PLOS Computational Biology

A sensitive model has high recall, while a specific model has high precision. To assess both qualities in tandem, the F1 score takes a harmonic mean of both such that only models that are both highly sensitive and specific achieve a high F1 score. This ensures that the metric is not biased towards the majority class.

The default classification threshold is set to 0.5, with higher or equal values classified as MRSA and lower values classified as MSSA. To assess the stability of model performance when this threshold is adjusted, a precision-recall curve is plotted, and the area under which is measured (AUPRC). A value approaching 1 signifies a stable model with non-random outputs.

5-fold cross-validation was performed on top-performing variants of each model type after hyperparameter tuning. All metrics were averaged across 5 folds and the best models were selected as final for clinical prospective testing.

For the prospective testing phase, selected top models were hosted privately for inference. The models were accessed via a Streamlit web interface in a read-only status during the prospective phase, ensuring that the data processing pipelines and the model parameters remained unchanged throughout this phase. The trained and validated models were tested on prospective samples using the AST results as the gold standard for comparison.

For the external validation phase, the same set of top models were used for inference. The model parameters remained unchanged, while modifications to the data processing pipelines were limited to those that adapt the pipeline to the different file format of the external dataset.

## Model confidence

Since all the tested model types output a continuous value in the range of 0–1, in addition to binary labels assignment, the output of the models may be interpreted as confidence in the prediction if model outputs are well-correlated with classification accuracy. Using a classification threshold of 0.5, where outputs larger than or equal to 0.5 are classified as MRSA, the confidence of the model in a prediction can be approximated as follows:

$$Confidence = |Output - 0.5| + 0.5$$

When predictions with higher confidence are more likely to be correct, a confidence threshold can be set in which only predictions with confidence higher than the threshold are accepted, while the rest are rejected. As a result, rejecting a small portion of outputs may increase the accuracy of the remaining predictions.

To investigate the utility of model confidence and enable comparison between models, confidence reliability plots and accuracy-rejection curves are examined for each model. Confidence reliability plots bin model outputs by confidence and show the accuracy for each bin. A well-calibrated model should have a linear correlation between model confidence and accuracy, such that model confidence reflects the probability of a prediction being correct. Accuracy-rejection curves show the extent of accuracy improvement when a percentage of low-confidence outputs is rejected. An ideal model should achieve high accuracy while rejecting the fewest outputs [19].

## Results

### Model structure

Grid search with manual hyperparameter adjustment was performed on all model types to optimize performance. Regarding the neural network, grid search results favored deep models with a large size of up to 120 million trainable parameters. The model was able to achieve good performance with a large dataset size and early stopping at minimum validation loss. Regarding other non-deep learning models, there was little to no change to the hyperparameters when compared with defaults or with the training attempts in the work of Weis *et al* [14].

## Model performance

After an initial round of hyperparameter tuning, the densely connected neural network, and boosting algorithms, including LightGBM, XGBoost, and CatBoost, were found to be superior. Different combinations of these models were subsequently tested to produce an ensemble model, in which a weighted average of the outputs of the neural network, LightGBM, and CatBoost in the ratio of 13:12:7 produced the optimal result. However, the addition of CatBoost to the ensemble significantly increased training and inference time, thus CatBoost was later removed from the ensemble with only a 0.03% absolute loss in accuracy, while the neural network and LightGBM remained in the ratio of 13:12. In the end, the ensemble model achieved an accuracy of 0.9442 (95% CI: 0.9432-0.9452) and AUPRC of 0.9872 during cross-validation, outperforming other models in the validation phase (Table 1).

Balancing model performance, variety, and computation time, the neural network, LightGBM, and the ensemble model were selected for prospective testing. A decrease in accuracy between 0.31% to 1.21% was observed when compared with the validation phase, but model performance remained stable in general. The three selected models showed very similar performance, with the ensemble model attaining the highest AUPRC at 0.9866 (Fig 2).

In external validation, all three models showed decrease in performance, but were still able to maintain specificity for MRSA predictions. The neural network, Light GBM and ensemble models achieved accuracy of 0.723, 0.695 and 0.722, respectively, and AUPRC of 0.8409, 0.8598 and 0.8765, respectively (Table 1 and Fig 3). Notably, there was a disproportionate increase in false negative predictions when compared to false positives. Therefore, it can be seen in Table 1 that all three models had high precision but low recall.

## Model confidence

During the validation phase, we observed two properties in top-performing models that support the use of confidence thresholds. First, the majority of model outputs had high confidence (Fig 4). This allowed a high confidence threshold to be set while still accepting most of the model predictions. Second, model confidence derived from the formula above

**Table 1. Model performance on 5-fold cross-validation, prospective testing and external validation. (AUPRC = area under the precision-recall curve).**

| Model | Accuracy | 95% CI | Precision | Recall | AUPRC |
|---|---|---|---|---|---|
| **5-fold cross-validation** | | | | | |
| Neural network | 0.9405 | (0.9386, 0.9424) | 0.9554 | 0.9380 | 0.9856 |
| LightGBM | 0.9419 | (0.9405, 0.9432) | 0.9587 | 0.9371 | 0.9859 |
| XGBoost | 0.9400 | (0.9390, 0.9407) | 0.9582 | 0.9341 | 0.9840 |
| CatBoost | 0.9417 | (0.9405, 0.9430) | 0.9594 | 0.9359 | 0.9853 |
| Random Forest | 0.9335 | (0.9315, 0.9355) | 0.9582 | 0.9220 | 0.9798 |
| Logistic Regression | 0.9240 | (0.9230, 0.9254) | 0.9392 | 0.9248 | 0.9786 |
| Support Vector Machine | 0.7120 | (0.6861, 0.7293) | 0.7092 | 0.8279 | 0.8475 |
| Ensemble | 0.9442 | (0.9432, 0.9452) | 0.9614 | 0.9385 | 0.9872 |
| **Prospective testing** | | | | | |
| Neural network | 0.9284 | (0.9185, 0.9374) | 0.9422 | 0.9438 | 0.9843 |
| LightGBM | 0.9388 | (0.9296, 0.9472) | 0.9557 | 0.9464 | 0.9849 |
| Ensemble | 0.9378 | (0.9285, 0.9462) | 0.9541 | 0.9464 | 0.9866 |
| **External validation** | | | | | |
| Neural network | 0.723 | (0.6941, 0.7505) | 0.8832 | 0.514 | 0.8409 |
| LightGBM | 0.695 | (0.6654, 0.7234) | 0.8736 | 0.456 | 0.8598 |
| Ensemble | 0.722 | (0.6931, 0.7496) | 0.8964 | 0.502 | 0.8765 |

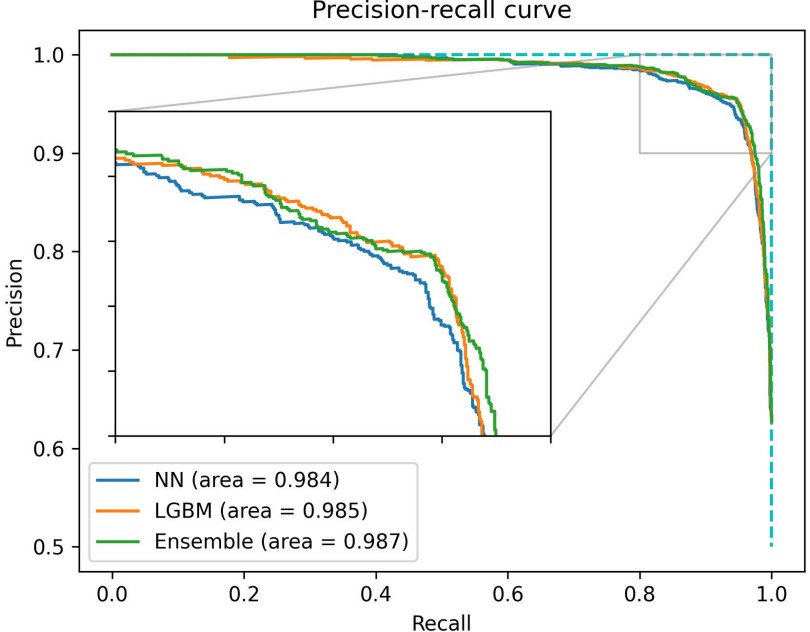

**Fig 2. Precision-recall curve of the neural network, LightGBM and ensemble models in the prospective testing phase, showing similar overall performance.**

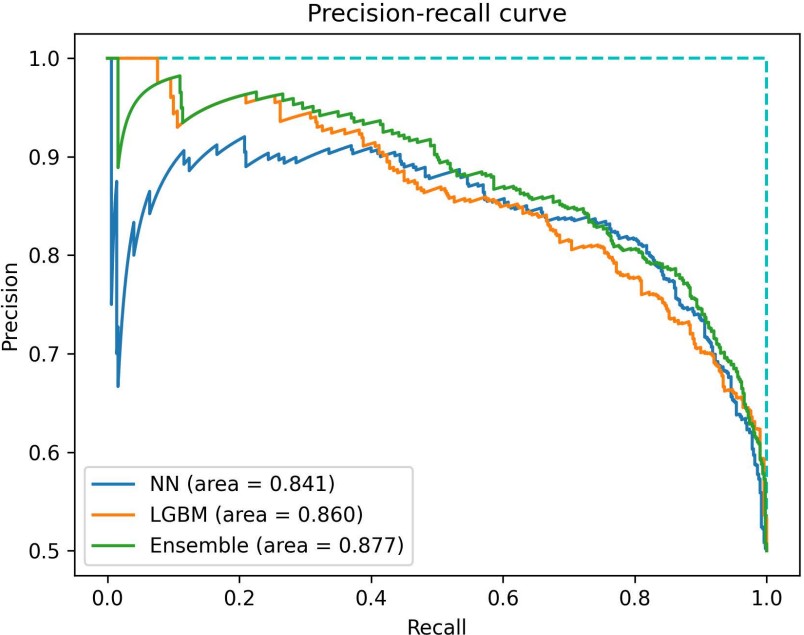

**Fig 3. Precision-recall curve of the neural network, LightGBM and ensemble models in the external testing phase.**

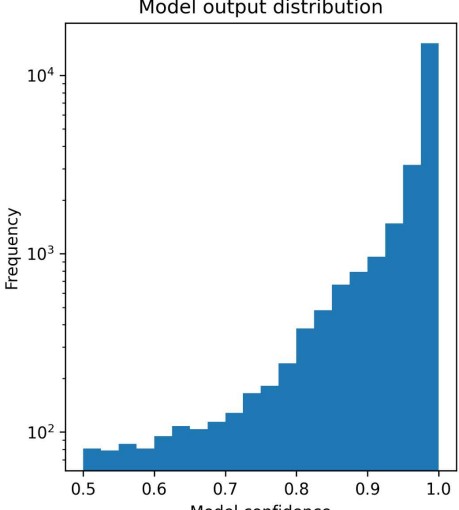
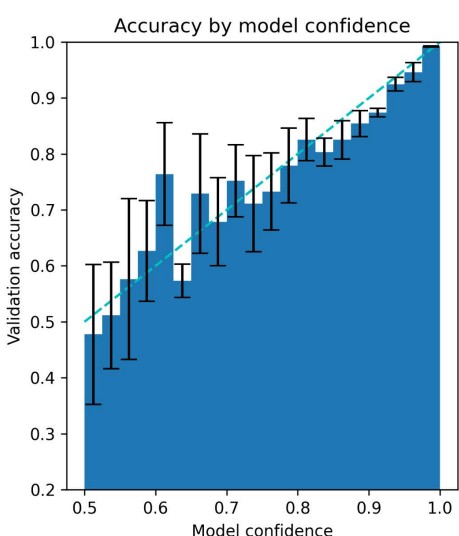

**Fig 4. Output analysis of the ensemble model during 5-fold cross-validation.** Left: distribution of model output summed across all folds. (Note: y-axis in logarithmic scale) Right: Confidence reliability shown as validation accuracy of each confidence interval. A well-calibrated model should follow the gray line indicating linear relationship between model confidence and accuracy. Error bars indicate 95% confidence interval from 5-fold cross-validation.

was highly correlated with prediction accuracy. Predictions were near-random at low model confidence (0.5) and had up to 0.98 accuracy at high confidence (1.0) (Fig 4). This indicates that the models are well-calibrated and that model confidence is a good estimation of the true probability of a prediction being correct.

The effect of confidence thresholds was evaluated in the prospective testing phase with accuracy-rejection curves. When all predictions were accepted, the three models had a raw accuracy between 0.9284 to 0.9388. When a threshold was applied, all models saw a steep increase in the accuracy of accepted predictions up to 20% sample rejection. When 20% of predictions were rejected, the neural network, LightGBM, and ensemble models achieved 0.9697, 0.9727, and 0.9777 accuracy respectively in the accepted predictions (Fig 5). In practice, rejected cases will not be reported in our laboratory and we will instead await the results of culture and sensitivity testing, which remains the gold standard method.

In the external validation, rejecting low confidence predictions also led to increased model accuracy, but the effect was less pronounced. As seen in Fig 6, the relationship between percentage of rejected samples and accuracy was linear. This may be attributed to many new resistant patterns in the external dataset that were not present in Hong Kong, such that a low percentage of sample rejection failed to exclude those patterns.

## Feature importance

To enhance the explainability of machine learning models, Shapley additive explanations (SHAP) were developed to visualize the importance of input features in model outputs. In this study, SHAP values were computed for the neural network, LightGBM, and the ensemble model, implemented using appropriate explainers in the shap Python package. The SHAP values of an ensemble model are computed using a weighted average of the SHAP values of its constituent models. Due to computational constraints, a random 1500-sample subset of the validation set was used to compute the SHAP values.

Feature importance of each model, presented as spectral plots, are available in S1 Fig. 3 common groups of important SHAP features are seen in all 3 models and are consistent across cross-validation folds when peak shifting has been accounted for. These are feature 999–1002 (m/z 4997–5009), feature 1526–1528 (m/z 6578–6584), and feature 1172–1175 (m/z 5516–5525), but the neural network and LightGBM differed substantially in the less significant features (S1 Fig).

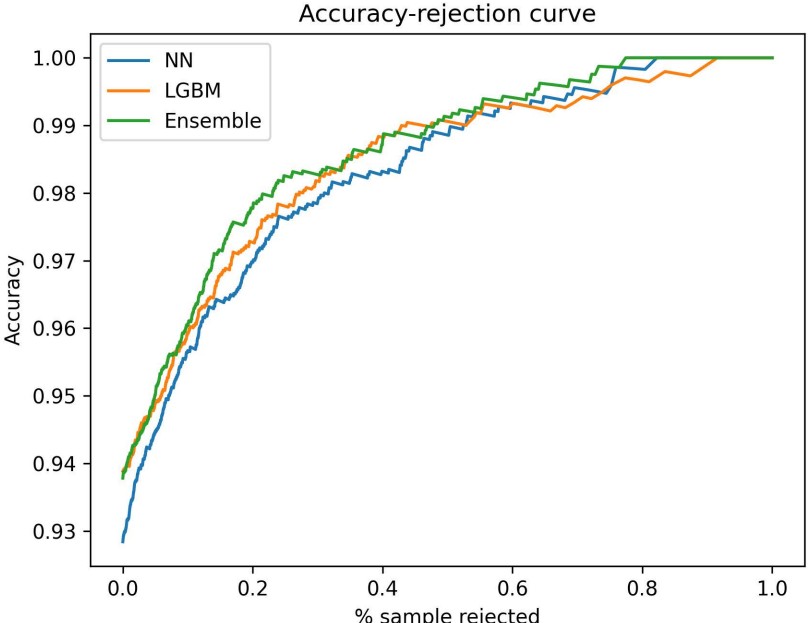

**Fig 5. Accuracy-rejection curves for all models in prospective testing.** The curves show steep incline between 0 to 20% sample rejection, with the ensemble model achieving the highest accuracy at 20% sample rejection.

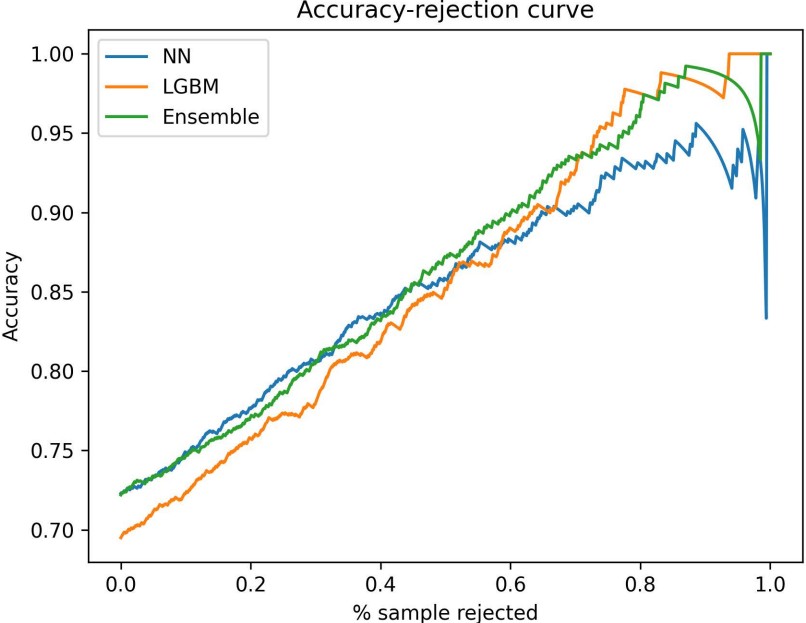

**Fig 6. Accuracy-rejection curves for all models in external validation.**

Feature 1526–1528 (m/z 6578–6584) and feature 1172–1175 (m/z 5516–5525) are consistent with potential markers found in previous studies, with minor position shift possibly attributed to the data preprocessing method used [3,20–23].

To further investigate the top 3 feature groups, pseudogel plots centered around the top feature groups were prepared using a random 500-sample subset of training data. With reference to Yu *et al.*, the pseudogel plot is a visualization tool for a large amount of spectral data to compare signal differences between a batch of MRSA and MSSA samples [3]. Each spectrum is represented by a horizontal row where the x-axis represents features, and the normalized intensity of each feature is represented by a color scale. 500 of such rows are stacked vertically to form the y-axis. A narrow band of difference in signal intensity can be seen in all three locations (S2 Fig).

However, it is important to note that no single feature group provides consistent discrimination between MRSA and MSSA. The relationships between top feature groups are illustrated with scatter plots in Fig 7, where each sample in the validation dataset is plotted on the graph based on the amplitude of the two feature groups. Although not completely consistent, it can be observed that the 999–1002 and 1526–1528 feature groups separated the samples into two distinct MRSA clusters and an undifferentiated mix, while the 1172–1175 feature group formed a MSSA cluster. We can infer that each feature group may be an independent predictor for a small group of samples, and accurate prediction in general can only be achieved when all predictors are accurately captured. This can be further illustrated by isolating these three feature groups in training. A multi-layer perceptron binary classifier trained on these three feature groups alone achieved an accuracy of 0.8107, signifying that these top feature groups covered most samples but did not capture all the necessary information to reach the high prediction accuracy of the proposed machine learning models.

To investigate the increase in false negative predictions during the external validation, pseudogel plots are prepared for the external dataset comparing MSSA samples, true positive MRSA samples, where the ensemble model predicts MRSA, and false negative MRSA samples, where the ensemble model predicts MSSA. 200 random samples are plotted for each category and shown in Fig 8, using the 1526–1528 feature group as an example. The 1526–1528 feature group is one of the top feature groups in the validation dataset as determined by SHAP values, but the plot shows no visible difference between MSSA and false-negative MRSA samples, while true-positive MRSA samples show visible increase in intensity

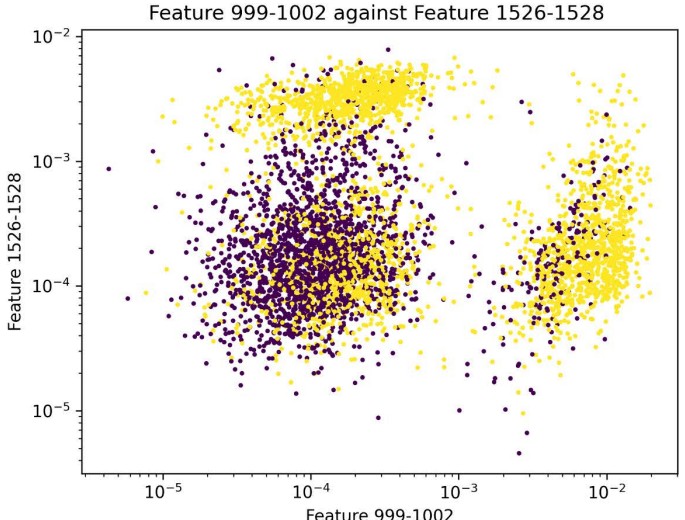
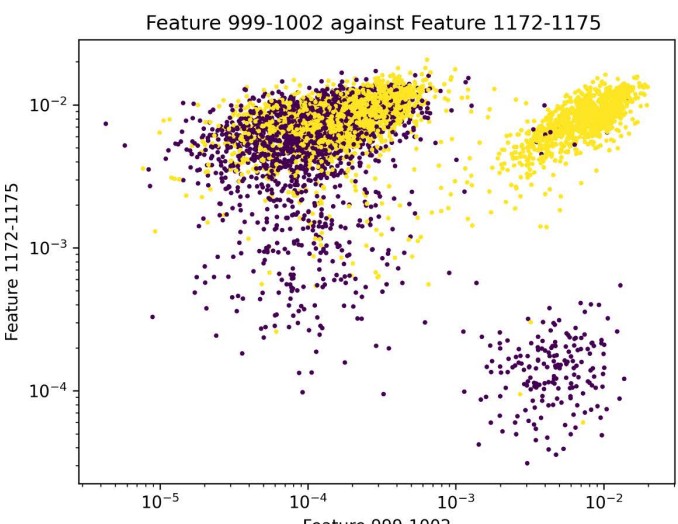

**Fig 7. Scatter plots using the validation dataset.** (Yellow: MRSA, purple: MSSA) Left: feature 999-1002 (m/z 4997-5009) against feature 1526-1528 (m/z 6578-6584). Right: feature 999-1002 (m/z 4997-5009) against feature 1172-1175 (m/z 5516-5525). (Note: axes in logarithmic scale) Although not completely consistent, the plots show feature 999-1002 and feature 1526-1528 forming two MRSA clusters and feature 1172-1175 forming a MSSA cluster.

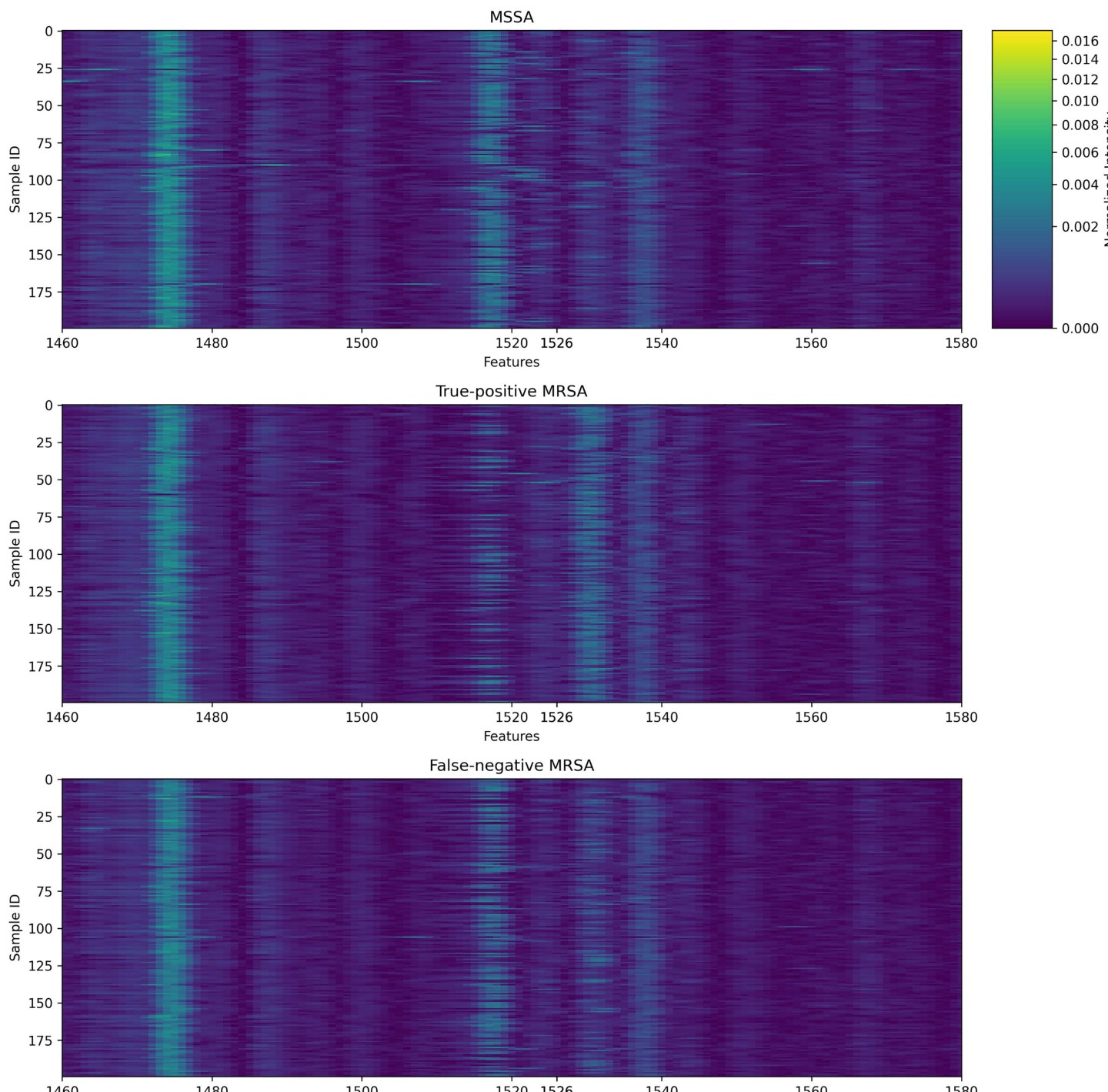

**Fig 8. Pseudogel plot of 200 MSSA samples, 200 true-positive MRSA samples (predicted to be MRSA by the ensemble model), and 200 false-negative MRSA samples (predicted to be MSSA by the ensemble model).** The plot is centered around the 1526-1528 feature group which was one of the top feature groups in the validation dataset. True-positive MRSA samples show visible differences from MSSA samples, but false-negative MRSA samples do not.

near feature 1526. This suggests that the model can recognize MRSA samples with similar intensity patterns as those from Hong Kong, thus maintaining high precision in the external dataset, but fails to generalize to unseen patterns, where the location of important peaks may be different.

## Discussion

There have been multiple attempts at antibiotic susceptibility prediction by extracting MALDI-TOF MS data using machine learning methods. However, multiple factors limited the model performance and clinical applicability of these models in previous studies, such as the use of additional laboratory processes, small model sizes, and small sample sizes (Table 2). In this study, we attempted to combat these factors with a large dataset and novel machine-learning techniques, and enhance model safety with confidence thresholds.

Sample size, confidence threshold and the geographical distribution of samples were found to be the biggest determinant of model performance. No single peak or pattern exists to enable reliable prediction of antibiotic susceptibility status in spectral data produced by an unseen lineage of *S. aureus*. Therefore, it is crucial that the training data encompasses a diverse set of samples to cover less commonly seen strains in the target geographical area. The confidence threshold provides additional safety but is not equally effective in all models. Neural networks with a logarithmic loss function seemed to be better calibrated than gradient boosting frameworks, with a better correlation between model confidence and accuracy. This may explain the increase in accuracy when ensembled with LightGBM despite LightGBM having better raw performance. Despite the good correlation, we acknowledge that further calibration, such as applying Platt scaling or temperature scaling, may further increase the effectiveness of confidence intervals and warrant further experimentation.

Despite the large sample size, all training samples used in this study were from the same regional laboratory covering two acute hospitals serving a population of 1 million people, which leads to a model that is geographically specific and has decreased performance in external validation. Close inspection of model performance on the external dataset suggests that classification of antibiotic susceptibility may be intrinsically not generalizable to other geographical areas without corresponding samples. Thus, safe application of such models should involve limiting the use of the models to the geographical areas they were trained in, and falling back to traditional AST methods for rejected low-confidence predictions. Further research is required to find a balance between prediction performance and wider geographical support.

Performance decay over time is also a concern as bacterial strains evolve. To investigate this effect, the neural network, LightGBM and ensemble models were trained using samples from 2021 and their performance was assessed by year. There was no significant trend in the performance per year, but the specimen collection time frame was too short to investigate the effect of bacterial strain turnover, thus effects in longer time frames are still unknown (S3 Fig).

Table 2. Comparison with studies from the previous 5 years. Studies investigating additional laboratory procedures before MALDI-TOF processing are excluded. Total sample size includes all S. aureus samples used for training, validation, and testing. Performance metrics are extracted from the validation phase of each study. (AUROC = area under the receiver operating characteristic curve).

| Study | Model type | Total sample size | Accuracy | AUROC |
|---|---|---|---|---|
| This study | Ensemble | 28462 | 0.9440 | 0.9838 |
| Wang et al., 2023 [8] | XGBoost | 31807 | – | 0.9358 |
| Yu et al., 2022 [3] | LightGBM | 20359 | – | 0.91 |
| Weis et al., 2022 [14] | LightGBM | 9000 (approx.) | – | 0.80 |
| Jeon et al., 2022 [4] | AMRQuest v2.1 | 553 | 0.876 | 0.876 |
| Kong et al., 2022 [2] | Random Forest | 548 | 0.7637 | 0.8244 |
| Chung et al., 2021 [11] | Random Forest | 25217 | 0.8142 | 0.8117 |
| Wang et al., 2021 [10] | Random Forest | 4858 | 0.8158 | 0.8997 |
| Tang et al., 2021 [24] | Genetic Algorithm | 214 | 0.8889 | – |

From a practical perspective, the neural network is a large model with over 120 million parameters. The time required to train this model without hardware acceleration is prohibitively long, but the training time can be massively shortened with hardware acceleration. A pre-trained version is also able to predict in real-time using CPU inference despite higher memory usage. From our experience, the neural network can predict 1000 samples in less than 1.5 seconds without hardware acceleration, with 3GB of memory usage. Comparatively, the LightGBM model has a smaller file size and less stringent hardware requirements. In view of training and memory cost, it may be reasonable in certain situations to forego the performance gains of the ensemble model in favor of a locally hosted LightGBM model.

In conclusion, we present a highly accurate model trained and validated on a large pool of samples, utilizing the same spectra generated in the species identification process with no additional manual procedures and cost implications. We also employ a novel parameter, namely confidence thresholds, in our model to further increase the accuracy by rejecting potentially inaccurate predictions. Finally, we develop a web interface to automate the data processing and prediction processes and enable batch prediction of multiple samples. Ease of use and high performance enables the potential for seamless integration into existing clinical workflows with a high degree of safety. However, the models are not directly generalizable across different geographical areas.

## Supporting information

**S1 Table. Base dataset characteristics.**
(DOCX)

**S2 Table. Class distribution across datasets used in this study.**
(DOCX)

**S1 Fig. Feature importance of the neural network, LightGBM, and ensemble model.**
(DOCX)

**S2 Fig. Pseudogel plot using a 500-sample random subset of training data.** A: feature 999–1002 (m/z 4997–5009), B: feature 1526–1528 (m/z 6578–6584), C: feature 1172–1175 (m/z 5516–5525).
(DOCX)

**S3 Fig. Accuracy by year of the neural network, LightGBM and ensemble model when trained on samples from 2021 only.**
(DOCX)

## Author contributions

**Conceptualization:** River Chun-wai Wong, Ni Tien, Christopher Koon-Chi Lai.

**Data curation:** Yik-Shun Lin, River Chun-wai Wong, Jiaxin Yu, Leo Chun-Hei Wong, Bang-Jau You.

**Formal analysis:** Yik-Shun Lin, River Chun-wai Wong, Kaichuang Yang, Christopher Koon-Chi Lai.

**Funding acquisition:** Christopher Koon-Chi Lai.

**Investigation:** Yik-Shun Lin, River Chun-wai Wong, Kaichuang Yang.

**Methodology:** Yik-Shun Lin, River Chun-wai Wong, Kaichuang Yang.

**Project administration:** Christopher Koon-Chi Lai.

**Resources:** Christopher Koon-Chi Lai.

**Software:** Yik-Shun Lin.

**Supervision:** Viola Chi-Ying Chow, Christopher Koon-Chi Lai, Margaret Ip.

**Validation:** Yik-Shun Lin, River Chun-wai Wong, Jiaxin Yu, Ni Tien, Bang-Jau You.

**Visualization:** Yik-Shun Lin.

**Writing – original draft:** Yik-Shun Lin, River Chun-wai Wong, Ho-Fung Leung.

**Writing – review & editing:** River Chun-wai Wong, Ingrid Yu-Ying Cheung, Viola Chi-Ying Chow, Christopher Koon-Chi Lai.

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
