## [Decision Letter · Decision Letter 0]

23 Sep 2025

PCOMPBIOL-D-25-01013

Application of Machine Learning with MALDI-TOF MS for Rapid Differentiation between Methicillin-susceptible and Methicillin-resistant Staphylococcus aureus

PLOS Computational Biology

Dear Dr. Lai,

Thank you for submitting your manuscript to PLOS Computational Biology. After careful consideration, we feel that it has merit but does not fully meet PLOS Computational Biology's publication criteria as it currently stands. Therefore, we invite you to submit a revised version of the manuscript that addresses the points raised during the review process.

Please submit your revised manuscript within 60 days Nov 23 2025 11:59PM. If you will need more time than this to complete your revisions, please reply to this message or contact the journal office at ploscompbiol@plos.org. Please include the following items when submitting your revised manuscript:

We look forward to receiving your revised manuscript.

Kind regards,

Alberto J M Martin, Ph.D.

Academic Editor

PLOS Computational Biology

James Faeder

Section Editor

PLOS Computational Biology

**Additional Editor Comments :**

Reviewer #1:

Reviewer #2: Address these comments

Reviewer #3: All 8 major issues raised by this reviewer should be addressed, specially those regarding the generalization capabilities of your tool

**Journal Requirements:**

1) Please upload all main figures as separate Figure files in .tif or .eps format. For more information about how to convert and format your figure files please see our guidelines:

2) Please ensure that all Figure files have corresponding citations and legends within the manuscript. Currently, Figures 3, and 5 in your submission file inventory do not have in-text citations. Please include the in-text citations of the figures.

3) We notice that your supplementary Figures, and Table are included in the manuscript file. Please remove them and upload them with the file type 'Supporting Information'. Please ensure that each Supporting Information file has a legend listed in the manuscript after the references list.

4) We note that the Data Availability Statement mentioned in the manuscript is different from that provided in the online submission form. The Data Availability statement in the online submission form is currently as follows: 'All relevant data are within the manuscript and its Supporting Information files.' While the one in the manuscript states "Datasets are not publicly available. All code used for model fitting, evaluation and

plotting, along with all performance metrics and plots are available on a GitHub repository at Please provide the complete Data Availability statement in the submission form and ensure that it matches the one mentioned in the manuscript.

Note:  Authors must share the “minimal data set” for their submission. PLOS defines the minimal data set to consist of the data required to replicate all study findings reported in the article, as well as related metadata and methods (https://journals.plos.org/plosone/s/data-availability#loc-minimal-data-set-definition).

5) Please provide a completed 'Competing Interests' statement, including any COIs declared by your co-authors. If you have no competing interests to declare, please state "The authors have declared that no competing interests exist". Otherwise please declare all competing interests beginning with the statement "I have read the journal's policy and the authors of this manuscript have the following competing interests:".

**Reviewers' comments:**

Reviewer's Responses to Questions

Reviewer #1: This study investigates the potential application of novel machine learning techniques in conjunction with MALDI-TOF MS to develop a predictive model capable of rapidly distinguishing between methicillin-susceptible Staphylococcus aureus (MSSA) and methicillin-resistant Staphylococcus aureus (MRSA). Such differentiation can facilitate more appropriate antibiotic use, improve treatment outcomes, and support antibiotic stewardship efforts.

The top-performing models included a large-scale neural network (NN), the LightGBM gradient boosting framework (LGBM), and a weighted ensemble combining the NN and LGBM models. All three models demonstrated strong classification accuracies(>0.92), which were further enhanced through the application of a confidence threshold.

Upon thorough assessment of this manuscript, I am pleased to confirm that my concerns regarding scientific justification, methodological transparency, reproducibility, and data interpretation have been thoroughly addressed. The manuscript presents a well-structured and scientifically sound study, with text and figure clarity, proper citations, detailed methodological descriptions, and enhanced visual materials.

I recommend that the manuscript be accepted for publication.

Reviewer #2: This paper considers an application of ML to MSRA classification. The interest in the contents is likely moderate for readers of PLOS Comp Bio. The application of ML is reasonable, and the reference are up-to-date. The method appears competitive with many modern methods.

My main concern is the use of confidence in this paper. The authors define confidence as = | − 0.5| + 0.5. This is not a statistical confidence measure, which the authors acknowledge by saying that it “can be interpreted as a confidence”. While it seems reasonable as a tool for calculating when to accept the prediction, based on Fig 2, I feel that the term confidence is likely not useful in this context. (However, the authors also define CIs for the models. Presumably, this is unrelated and calculated based on the confusion matrix itself.)

Having said this, some standardization of the “confidence” would be helpful. I believe the authors would benefit from using the literature on “machine learning with rejection”. For instance, this would allow the authors to standardize whether they refer to this condition as being “ignored” (line 422) or “rejected” (later on line 422). The authors may also benefit from the measures in the literature that discuss evaluation in the presence of rejection. An appropriate survey would be Hendrickx et al. 2024 (https://link.springer.com/article/10.1007/s10994-024-06534-x ). I would encourage the authors to include some of these additional evaluation criteria (beyond what is shown in Fig 3) to help the readers understand the impact of rejecting, e.g., 20% of inputs.

There is slight imbalance in the data set but the authors have appropriately used classification metrics that indicate that the class imbalance does not falsely inflate the results. Presumably, when the authors do their train-test and validation splits, they are done in a stratified manner. Please confirm or give data on the variation in data imbalance in the validation and prospective test sets.

The authors note on lines 115-117 “Limiting the scope of sample collection may potentially enhance model accuracy by focusing on features of local strains.” I believe this is intended to be described as an avenue for improving results, but I feel that readers may also want to consider that this offers an avenue for improving generalizability: that models that are insensitive to geographical location may be the goal of future work (or, on the other hand, that such models are unlikely to ever exist). Some more material here (or in another suitable location in the paper) would be beneficial.

Line 120 – reference is missing to Yu et al paper.

Reviewer #3: Summary: This paper applies machine learning to classify MRSA versus MSSA using MALDI-TOF spectra. The dataset is large (24,487 retrospective isolates and 2,975 prospective isolates from a single hospital), and models include a neural network, LightGBM, CatBoost, XGBoost, and an ensemble. Reported accuracies are 0.928–0.939 without thresholding, rising to >0.97 when ~20% of low-confidence cases are rejected. The study also incorporates SHAP-based feature analysis and a Streamlit web application for prospective testing. The dataset size and prospective validation are notable strengths. However, there are major concerns with reproducibility, generalizability, and several aspects of the methodology. At its current stage, the manuscript does not meet the journal’s standards.

Major Critiques:

1. Methods (p.17, Lines 305-311): All isolates are from a single hospital in Hong Kong. Without external validation on other sites or instruments, the results cannot be generalized. The authors should provide such data to verify their model’s generalizability.

2. Discussion (p.16, Lines 283–290): The neural network requires hardware acceleration, while LightGBM is lightweight. The authors do not quantify model size or inference speed. Reporting accuracy versus inference time and memory is essential. Smaller NN architectures should be explored.

3. Results (p.11, Lines 168–190): Accuracy gains rely on rejecting up to 20% of cases. The manuscript does not explain how rejected cases would be managed in practice or why rejection is preferable to other strategies. This undermines the clinical relevance of the approach.

4. Methods (p.19, Lines 368–373): The same preprocessing pipeline is applied across all folds. Please clarify if spectra from the same patient or episode could appear in both train and test sets, and how deduplication was enforced. This is critical for unbiased validation.

5. Results (p.12–13, Lines 195–238): SHAP identifies feature groups, but scatterplots show limited separation. Claims about consistent discriminatory power should be tempered. Consider reporting classification accuracy using only top SHAP features to strengthen the interpretation.

6. Discussion (p.15, Lines 277–281; Supp. Fig. 3): The four-year analysis shows no clear decay, but this horizon is short. Bacterial strain turnover often requires longer monitoring. The authors should acknowledge this limitation.

7. Results (p.14, Lines 260–263; Table 2): The authors state sample size and confidence threshold are the biggest determinants of accuracy. However, Table 2 shows larger datasets with lower performance. This inconsistency suggests other factors matter.

8. Data and code availability (p.23, Lines 447–450): The datasets are not publicly available. Without this, the study is not reproducible.

Minor Critiques:

1) Abstract (p.3, Lines 33–59): The abstract is results-heavy. Streamline to emphasize clinical motivation, dataset size, and key findings.

2) Discussion (Table 2, p.14-15): Weis et al. (Nat Med 2022) is a major benchmark using the DRIAMS dataset (four hospitals, cross-site validation). This study is only briefly mentioned. A deeper comparison is needed, highlighting that Weis achieved generalization across sites, whereas this work is limited to one site.

3) Discussion (p.16, Lines 288–290): Report prediction times for NN, LightGBM, and ensemble models. Practical feasibility depends not just on accuracy but also speed.

4) Supp. Table 1 (p.29, Lines 577-579): How MRSA/MSSA ratios differ across specimen types is missing. Furthermore, clarify whether class weighting or balancing was used to avoid bias.

5) Methods (p.21, Lines 404–408): Clarify whether preprocessing and inference were frozen before prospective testing. Pipelines should be locked prior to deployment.

6) Confidence–accuracy plots (Fig. 3): Add error bars or confidence intervals to illustrate variability across folds.

7) SHAP stability: State whether top SHAP features are consistent across folds or vary between runs.

Issue of Grammar, Spelling, Proofreading, etc.

1) Several sentences are long and would be clearer if split (e.g., Introduction, p.6, Lines 87-91).

2) Figure captions are short; expand them so each figure can be understood without referring to the main text.

**Have the authors made all data and (if applicable) computational code underlying the findings in their manuscript fully available?**

Reviewer #1: Yes

Reviewer #2: **No:** The authors state that data is available, but it is not, presumably for privacy reasons.

Reviewer #3: Yes

PLOS authors have the option to publish the peer review history of their article (what does this mean?). If published, this will include your full peer review and any attached files.

Reviewer #1: No

Reviewer #2: No

Reviewer #3: No

**Figure resubmission:**
---

## [Decision Letter · Decision Letter 1]

17 Nov 2025

Dear Dr Lai,

We are pleased to inform you that your manuscript 'Application of Machine Learning with MALDI-TOF MS for Rapid Differentiation between Methicillin-susceptible and Methicillin-resistant Staphylococcus aureus' has been provisionally accepted for publication in PLOS Computational Biology.

Best regards,

Alberto J M Martin, Ph.D.

Academic Editor

PLOS Computational Biology

James Faeder

Section Editor

PLOS Computational Biology

Reviewer's Responses to Questions

**Comments to the Authors:**

Reviewer #1: The manuscript is scientifically robust, and the methodology is well-justified. In light of these strengths, I recommend that it be accepted for publication.

Reviewer #2: The authors have addressed the comments in my previous review. I believe the article should be accepted.

Reviewer #3: All the comments/concerns were addressed.

**Have the authors made all data and (if applicable) computational code underlying the findings in their manuscript fully available?**

Reviewer #1: Yes

Reviewer #2: Yes

Reviewer #3: None

PLOS authors have the option to publish the peer review history of their article (what does this mean?). If published, this will include your full peer review and any attached files.

Reviewer #1: No

Reviewer #2: No

Reviewer #3: **Yes:** Rajib Saha

---

## [Editor Report · Acceptance letter]

PCOMPBIOL-D-25-01013R1

Application of Machine Learning with MALDI-TOF MS for Rapid Differentiation between Methicillin-susceptible and Methicillin-resistant Staphylococcus aureus

Dear Dr Lai,

I am pleased to inform you that your manuscript has been formally accepted for publication in PLOS Computational Biology. Your manuscript is now with our production department and you will be notified of the publication date in due course.

With kind regards,

Zsofia Freund
